   

# Microplastics as environmental modifiers of lung disease

Emmanouela Epeslidou[1], Julia S Scott[1], Bim de Klein[1], Jeremy Tan Cudia[2], Barbro Melgert [ID][2,4]✉ & Stefan Prekovic [ID][1,3,4]✉

## Abstract

**Human-driven environmental change continues to reshape global patterns of disease, as seen in past pollution-related respiratory crises. Microplastics, persistent synthetic polymer particles, have now emerged as a widespread airborne contaminant with growing relevance for lung health. Continuous inhalation exposure, particularly in indoor environments rich in synthetic fibers, raises concern about their contribution to respiratory disease. Epidemiological and experimental studies increasingly link microplastic exposure to lung cancer, asthma, chronic obstructive pulmonary disease, and pulmonary fibrosis, yet the underlying mechanisms remain poorly defined. This review integrates current evidence on how particle properties influence biological outcomes and outlines how different polymer types, sizes, and aging states affect lung cells through inflammation, oxidative stress, ferroptosis, epithelial–mesenchymal transition, and epigenetic change. Together, these findings suggest that microplastics may act as environmental modifiers that exacerbate disease progression. Recognizing their complex and persistent nature highlights the need for standardized exposure metrics, mechanistic research at realistic doses, and coordinated scientific and regulatory action.**

**Keywords** Microplastics; Lung; Cancer; COPD; Asthma
**Subject Categories** Evolution & Ecology; Respiratory System

## Introduction

History clearly illustrates how human activities can redefine health trajectories through environmental changes. Take, for example, the rapid industrialization of nineteenth-century London; in just a few decades, this urban center transitioned into an industrial powerhouse, fueling growth through unprecedented coal consumption. Yet, this progress came at a severe cost: dense, persistent coal-smoke smog enveloped the city, turning day into twilight, staining buildings black, and profoundly altering the daily lives of Londoners. Respiratory disorders, previously uncommon, surged markedly, signaling a clear connection between environmental pollution and human disease emergence (Hanlon, 2015). This historical example underscores a critical lesson—human-induced environmental changes inevitably shape patterns of health and disease, often with far-reaching and unforeseen consequences. Today, another pervasive, yet less visible anthropogenic threat has emerged in the form of microplastics. These synthetic polymer particles (<5 mm diameter) have infiltrated ecosystems globally, highlighting their omnipresent nature (Cocca et al, 2023). Microplastics originate either from intentional manufacturing (*primary microplastics*), or from the breakdown of larger plastic debris and fibers shed from synthetic textiles (*secondary microplastics*). Due to their small size, these particles are readily inhaled or ingested, leading to continuous human exposure (Fig. 1). Recent studies demonstrate the systemic presence of microplastics, with particles detected in human blood, lung, heart, testes, and brain (L Leonard et al, 2024; Amato-Lourenço et al, 2021; Yang et al, 2023b; Zhao et al, 2023; Nihart et al, 2025), underscoring their capacity for widespread bioaccumulation within the human body. While the clinical consequences of this accumulation remain uncertain, their detection across multiple organ systems raises important concerns regarding potential long-term effects on human health.

Continuous inhalation exposure makes the lung particularly susceptible to airborne microplastics, especially in indoor settings where synthetic textile fibers dominate airborne particle populations (Amato-Lourenço et al, 2021; Vasse and Melgert, 2024). Concentrations indoors may surpass outdoor levels by at least fourfold, driven by routine activities such as doing laundry and the wearing of synthetic garments, thereby amplifying respiratory exposure given that individuals spend most of their time indoors (Eberhard et al, 2024). Occupational studies further support the clinical significance of inhaled microplastics, associating workplace exposure, particularly to polyvinyl chloride (PVC) and polyamide (nylon) fibers, with severe respiratory conditions, including lung cancer (Fig. 1) (Mastrangelo et al, 2003; Girardi et al, 2022; Özgen Alpaydin et al, 2024; Kern et al, 1998; Turcotte et al, 2013). Despite the emerging epidemiological evidence, the molecular mechanisms underpinning microplastic-induced lung pathology remain poorly defined.

This review integrates epidemiological insights and mechanistic evidence to position inhaled microplastics as environmental modifiers that may be reshaping lung disease trajectories (Table 1), underscoring a critical yet underappreciated public health threat demanding scientific attention.

---

[1]University Medical Center Utrecht, Utrecht, The Netherlands. [2]University of Groningen, Groningen, The Netherlands. [3]Singapore Eye Research Institute, Singapore, Singapore. [4]These authors contributed equally as senior authors: Barbro Melgert, Stefan Prekovic. ✉E-mail: b.n.melgert@rug.nl; s.prekovic@umcutrecht.nl

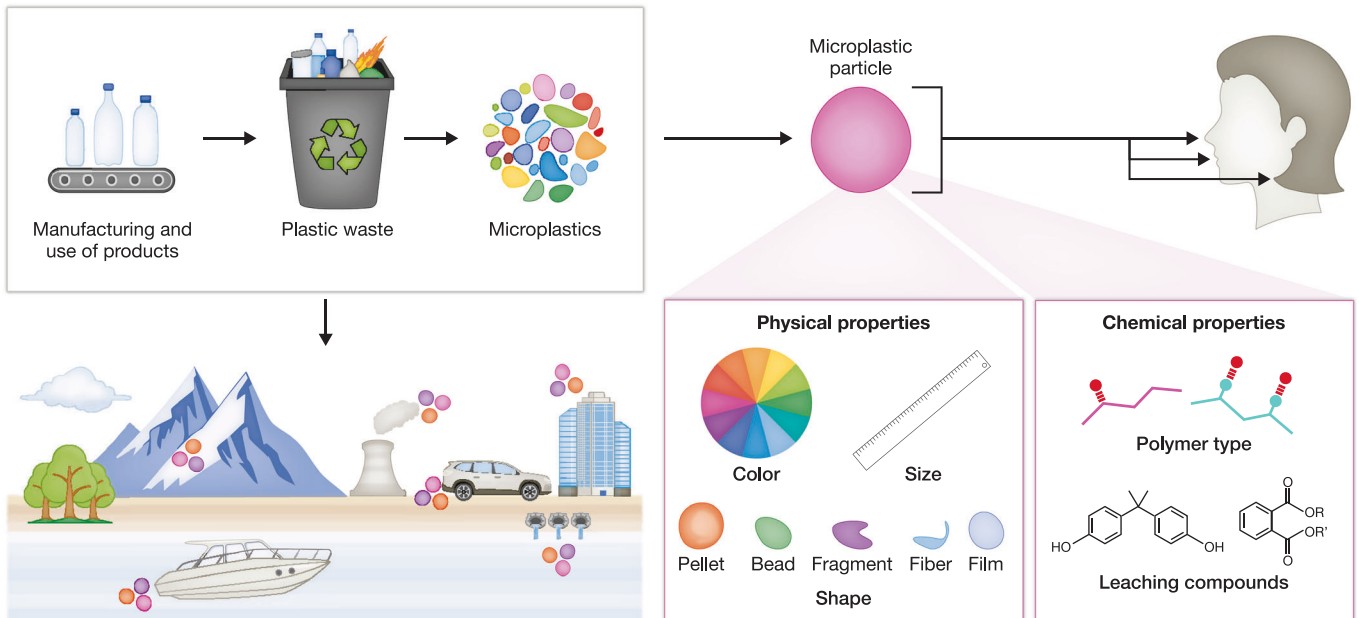

**Figure 1.  Schematic overview of microplastic sources, environmental distribution, and human exposure routes.**

Synthetic polymers from daily-use products degrade into microplastics, infiltrating ecosystems and accumulating within various environmental compartments. Microplastic particles exhibit distinct physical properties (color, size, and shape) and chemical characteristics (polymer composition and additive leaching), critically determining their environmental fate and potential toxicity. Humans are primarily exposed via inhalation, ingestion, and dermal contact, highlighting the importance of understanding particle properties to assess health risks accurately.

# What are microplastics?

Microplastics constitute a diverse class of synthetic polymer particles derived primarily from petroleum-based fossil fuels (Geyer et al, 2017). Their prevalence in the environment results from the extensive use of plastics, whose durability, versatility, and low cost underpins their pervasive integration into modern daily life (Andrady and Neal, 2009). However, the same properties make plastics exceptionally resistant to environmental degradation, allowing microplastics to persist in ecosystems for decades or possibly even millennia (Sonke et al, 2022), consistent with model projections showing continued cycling of small microplastics through Earth's surface reservoirs for several thousand years. Microplastics typically originate via two main pathways: (1) direct release of intentionally manufactured microscale particles (Fendall and Sewell, 2009; Cole et al, 2011) (e.g., *microbeads in personal care products and cleaning agents*) or (2) indirect formation through the fragmentation and abrasion of larger plastic debris (Andrady, 2011; Cole et al, 2011). Prominent sources include synthetic textile fibers released during laundering, degraded plastic waste, and manufactured microbeads (Cole et al, 2011), collectively contributing to their widespread ecological distribution and impact (Fig. 1). These particles, predominantly fibers and irregular fragments, have been detected across diverse ecosystems, underscoring their widespread presence and potential ecological consequences.

The environmental fate and biological impact of microplastics are determined by their physicochemical properties, notably particle size, density, color, shape, and polymer crystallinity. Particle size is especially influential, with larger microplastics (>10 μm) predominantly remaining in upper environmental compartments (*surface waters, soil surfaces, and atmospheric deposition zones*) and affecting organisms capable of size-specific ingestion (Cole et al, 2013; Wright et al, 2013), while smaller (<2.5 μm) and ultrafine particles (<0.1 μm) penetrate deeply into biological systems, significantly increasing their toxic potential (Prüst et al, 2020; Yong et al, 2020). However, accurate quantification of these smaller particles remains challenging, complicating exposure and risk assessments. Polymer density further shapes environmental distribution: lower-density polymers such as polypropylene (0.85–0.94 g/cm³) and polyethylene (0.92–0.97 g/cm³) tend to float or remain suspended in aquatic environments, primarily influencing pelagic and surface-dwelling organisms (Zhang et al, 2017; Hidalgo-Ruz et al, 2012), whereas higher-density polymers like polyvinyl chloride (1.38 g/cm³) readily sediment, directly impacting benthic organisms at sediment-water interfaces (Gomiero et al, 2018; Koelmans et al, 2017). Additionally, particle shape significantly modulates environmental mobility and biological interactions. For example, fibers have a high specific surface area, which exhibit enhanced pollutant adsorption capabilities and greater potential to interact with organisms (Zhao et al, 2022a). Conversely, pellets and irregular fragments differ in their behavior, typically displaying higher ingestion rates by aquatic organisms (Mato et al, 2001; Boettcher et al, 2023; Lusher et al, 2013; Li et al, 2016). Lastly, films are thin, with irregularities on the surface and have a tendency to fold, which results in a high maximum adsorption capacity and biofilm growth, potentially facilitating contaminant transfer within ecosystems (Rozman et al, 2023). Polymer crystallinity also substantially affects environmental degradation rates: semicrystalline polymers (e.g., polyamide) exhibit higher mechanical strength and resistance to degradation,

**Table 1. Summary of microplastics and nanoplastics discussed in this review categorized by size and associated cellular response in the context of health tissue or disease.**

| Source | Particle | Polymer class | Particle size (diameter) | Particle shape | Cellular response | Disease |
|---|---|---|---|---|---|---|
| Yang et al, 2023a | Polyethylene | Polyolefin, synthetic | 500 nm | Irregular[a] | Lysosomal damage and macrophage triggering, affected brain function through gut-brain axis (mouse model) | – |
| Wolff et al, 2023 | Polystyrene; PMMA | Vinyl polymer, synthetic; Acrylic polymer, synthetic | 50, 200, 1000 nm; 70, 400, 1100 nm | Spherical | Affected immune checkpoint marker expression (human immune cells in vitro) | – |
| Tavakolpournegari et al, 2024 | PET | Polyester, synthetic | 163–292 nm | Irregular[a] | Increased ROS production, macrophage polarization to M1 (murine macrophages in vitro) | – |
| Weber et al, 2022 | PVC | Vinyl polymer, synthetic | 50–600 nm | Irregular[a] | Inflammatory response (human immune cells in vitro) | – |
| Jin et al, 2024 | PVC | Vinyl polymer, synthetic | 6.5–25 μm | Spherical[a] | Induced senescence and increased ROS levels in lung epithelium (mouse model and human cells in vitro) | – |
| Aloi et al, 2024 | Polystyrene (photoaged) | Vinyl polymer, synthetic | 1–1.4 μm | Spherical | Reduced cell viability, increased ROS production, elevated DNA damage and inflammation (murine macrophages in vitro) | – |
| Merkley et al, 2022 | Polystyrene | Vinyl polymer, synthetic | 10 μm | – | Metabolic shift of macrophages towards glycolysis, reduction of mitochondrial respiration (murine macrophages in vitro) | – |
| Adler et al, 2024 | Polystyrene | Vinyl polymer, synthetic | 0.5, 1, 3 μm | Spherical[a] | Increased necrosis, ROS production, altered metabolic activity (human macrophages in vitro) | – |
| Weber et al, 2022 | Polystyrene | Vinyl polymer, synthetic | 50–600 nm | Irregular[a] | Inflammatory response (human immune cells in vitro) | – |
| Yang et al, 2021 | Polystyrene | Vinyl polymer, synthetic | 40 nm | Spherical[a] | Oxidative stress, inflammatory responses, epithelial barrier destruction, disruption of tight junctions (human lung epithelial cells in vitro) | – |
| Wu et al, 2024 | Polystyrene | Vinyl polymer, synthetic | 20 nm, 10 μm | Spherical[a] | Apoptosis, ferroptosis, endoplasmic reticulum stress in lung epithelium (mouse model and human cells in vitro) | – |
| Lin et al, 2022 | Polystyrene | Vinyl polymer, synthetic | 80 nm | Spherical[a] | Induction of mitochondrial dysfunction and metabolic toxicity pathways (human hepatic and human lung cells in vitro) | – |
| Yu et al, 2025 | Polystyrene | Vinyl polymer, synthetic | 5 μm | Spherical | Tight junctions and transcriptional regulation disruption, lung dysplasia, affected lung development (rat offsprings) | – |
| Luo et al, 2023 | Polystyrene | Vinyl polymer, synthetic | 100 nm | Spherical[a] | Cellular senescence, lung inflammation, lung dysfunction (rat model) | – |
| Vlacil et al, 2021 | Polystyrene | Vinyl polymer, synthetic | 1 μm | – | Enhanced inflammatory cytokine release, vascular inflammation, adhesion molecule expression (mouse model, murine myocardial endothelial and murine monocytic cells in vitro) | – |
| Song et al, 2024 | Nylon | Polyamide, synthetic | 1–5 μm, 5–10 μm | Irregular sphere-shaped fibers[a] | Impaired epithelial differentiation, maturation, regeneration and repair (human and murine alveolar and airway-type organoids in vitro) | – |
| Li et al, 2025b | PAN | Resin polymer, synthetic | 200–300 nm | Fibers[a] | Airway remodelling, EMT transition, cilia formation (mouse model and human lung cells in vitro) | Cancer |
| Traversa et al, 2024 | Polyethylene | Polyolefin, synthetic | 0.2–9.9 μm | Spherical | Migration and EMT transition (human lung cells in vitro) | Cancer |
| Rafazi et al, 2024 | Polyethylene | Polyolefin, synthetic | 37–75 μm | Polyhedral[a] | Increase cell proliferation and migration (human glioblastoma cells in vitro) | Cancer |

**Table 1.** (continued)

| Source | Particle | Polymer class | Particle size (diameter) | Particle shape | Cellular response | Disease |
|---|---|---|---|---|---|---|
| Brynzak-Schreiber et al, 2024 | Polystyrene | Vinyl polymer, synthetic | 0.25, 1, 10 μm | Spherical[a] | Increased cell migration (human colorectal cells in vitro) | Cancer |
| Goodman et al, 2021 | Polystyrene | Vinyl polymer, synthetic | 1, 10 μm | Spherical[a] | Enhanced cellular motility, development of filopodia and focal adhesions (human lung cells in vitro) | Cancer |
| Ernhofer et al, 2025 | Polystyrene | Vinyl polymer, synthetic | 0.24, 1 μm | Spherical | Increased internalization, migration, DNA damage, oxidative stress, and activation of survival pathways- early tumor promotion (human lung cells and organoids in vitro) | Cancer |
| Paplińska-Goryca et al, 2025 | Nylon | Polyamide, synthetic | 80, 160 μm | Fibers[a] | Upregulation of metabolic pathways, intensified inflammatory response, chronic inflammation (asthma patients); Increased cell motility, inflammatory recruitment, epithelial barrier disruption (in COPD patients) | Obstructive Lung disorders |
| Wang et al, 2025; Wei et al, 2024 | Polystyrene | Vinyl polymer, synthetic | 100 nm; 5–5.9 μm | Spherical; spherical | Pulmonary inflammation, epithelial barrier dysfunction, increased ROS production, ferroptosis and exacerbation of allergic asthma (asthma mouse model) | Obstructive Lung disorders |
| Yang et al, 2024; Wei et al, 2025 | Polystyrene | Vinyl polymer, synthetic | 40 nm; 2 μm | Spherical; spherical[a] | Neutrophilic inflammation, oxidative stress, ferroptosis, autophagy, disruption in alveolar architecture, mitochondrial dysfunction (mouse COPD model) | Obstructive Lung disorders |
| Xu et al, 2025 | Polystyrene | Vinyl polymer, synthetic | 200 nm, 2.5 μm | Spherical[a] | Apoptosis in airway epithelial cells, airway hyperresponsiveness and inflammation (asthma mouse model) | Obstructive Lung disorders |
| Han et al, 2023 | Polystyrene + DEHP | Vinyl polymer, synthetic + plasticizer | 0.2–0.5 μm | Spherical[a] | Oxidative stress, airway inflammation, exacerbation of allergic asthma (asthma mouse model) | Obstructive Lung disorders |
| Zhang et al, 2024 | Polystyrene | Vinyl polymer, synthetic | 5 μm | Spherical[a] | Ferroptosis in alveolar epithelial cells (mouse model and human cells in vitro) | Lung fibrosis |
| Li et al, 2022 | Polystyrene | Vinyl polymer, synthetic | 5 μm | Spherical[a] | Enhanced expression of fibrotic markers, oxidative stress, alveolar epithelial injuries (mouse model) | Lung fibrosis |
| Kang et al, 2024 | Polystyrene | Vinyl polymer, synthetic | 5 μm | Spherical[a] | Compromised epithelial barrier integrity, imbalance in iron metabolism, ferroptosis, lung injury (mouse model) | Lung fibrosis |

[a] Validated by group before use.

whereas amorphous polymers (e.g., PVC) degrade more readily due to structural flexibility and vulnerability to environmental stressors (Li et al, 2025a; Mendez et al, 2025; Shi et al, 2024). Environmental aging preferentially targets amorphous polymer regions, progressively diminishing mechanical integrity, altering surface chemistry, and thereby influencing interactions with environmental contaminants and organisms (Shi et al, 2024). In addition, color provides further insight into microplastic sources and aging status, with brightly colored, sharp-edged particles typically indicating recent introduction, while faded, transparent, or smooth-edged particles reflect prolonged oxidative weathering and photodegradation (Zhao et al, 2022b).

Chemically, microplastics comprise polymer matrices combined with numerous additives such as plasticizers (e.g., phthalates), antioxidants, stabilizers, flame retardants, and dyes, that are integrated during polymer production to support material properties (Wiesinger et al, 2021; Hahladakis et al, 2018; Monclús et al, 2025). These additives can readily leach into surrounding environments, a process influenced by polymer characteristics and environmental conditions. For instance, PVC, characterized by its high chlorine content (Lu et al, 2023), undergoes photodegradation upon prolonged UV exposure, releasing hazardous chlorinated compounds such as dioxins (Kudzin et al, 2023), which are recognized as persistent environmental pollutants known to disrupt hormonal signaling and induce severe potent toxic effects (White and Birnbaum, 2009; Birnbaum, 1995). Additive leaching rates differ significantly depending on chemical structure and polymer type. Plasticizers, particularly bisphenol A (BPA) and phthalates, exhibit notable leachability and have been widely detected in aquatic and terrestrial systems, raising substantial concerns about endocrine disruption and ecological impacts (Kumawat et al, 2022; Martínez-Ibarra et al, 2021). Furthermore, oxidative aging processes alter microplastic surface chemistry, notably through the formation of reactive functional groups (e.g., carbonyl groups), thereby increasing their affinity to bind environmental pollutants such as heavy metals, polycyclic aromatic hydrocarbons, and persistent organic pollutants (Chen et al, 2024; Binda et al, 2023). This enhanced binding capacity positions microplastics as efficient environmental vectors, significantly facilitating pollutant distribution, bioavailability, and toxicological risks across ecosystems.

In view of the abovementioned diversity of microplastics, it has become evident that a wide range of biological responses may arise even from particles that differ only in shape or surface state. Fibrous and elongated microplastic particles are taken up less efficiently by phagocytes than spherical ones, and therefore are more likely to trigger frustrated phagocytosis, leading to cellular responses that differ from those induced by compact particles (Vasse and Melgert, 2024; Wieland et al, 2022). Environmental aging further widens these differences as UV/photo-oxidation introduces oxygenated groups and surface defects, increasing co-contaminant adsorption and reactivity and leading to higher oxidative stress and stronger pro-inflammatory responses in macrophage models compared with pristine particles (Xu et al, 2024; Yu et al, 2024; Aloi et al, 2024). In parallel, mechanistic work indicates that microplastics can engage inflammasome pathways via lysosomal damage and ROS, particularly when surfaces are defect-rich or carry adsorbed pollutants (Alijagic et al, 2023). These observations indicate that microplastics are not a monolithic hazard, as shape and surface state meaningfully shift biological responses, yet only a limited number of studies have examined these physicochemical differences under controlled and comparable conditions, highlighting the need for systematic research to disentangle particle-specific from context-dependent effects.

Together, these distinctive physical and chemical attributes highlight microplastics as a fundamentally different class of pollutants compared to traditional environmental contaminants. Unlike conventional pollutants, which generally dissipate, degrade, or diminish in toxicity over time, microplastics composed of diverse polymeric materials persist for exceptionally long periods due to their inherent resistance to biodegradation. Their particulate nature and chemical complexity enable continuous interaction with biological systems and promote their accumulation, substantially enhancing toxic potential. Furthermore, the dual capacity of microplastics to adsorb and concentrate environmental toxins while simultaneously releasing embedded chemical additives distinguishes them uniquely among pollutants, creating a sustained and multifaceted toxicological threat. This complex mode of action underscores the necessity to approach microplastics not simply as another environmental pollutant, but as a distinct category of environmental hazard requiring dedicated analytical tools, specialized risk assessment frameworks, and targeted regulatory strategies.

## What are the exposure routes and how does exposure affect lung function?

Human exposure to microplastics occurs primarily through three distinct routes: ingestion, dermal contact, and inhalation. Among these pathways, dietary ingestion has received considerable attention due to widespread microplastic contamination of foods, particularly driven by bioaccumulation in marine and freshwater organisms (Giri et al, 2024). While dermal contact remains comparatively understudied and is generally mitigated by the skin's protective barrier, prolonged exposure or compromised skin integrity could facilitate microplastic penetration, especially through frequent use of personal care products containing synthetic microbeads or direct contact with contaminated surfaces (Sun and Wang, 2023). However, inhalation has emerged as a critical exposure route driven by elevated airborne microplastic concentrations in indoor environments, largely attributed to synthetic textiles, household dust, and limited ventilation (Eberhard et al, 2024). In fact, recent estimates suggest adults may inhale up to ~68,000 particles per day in the 1–10 μm size range in indoor settings (Yakovenko et al, 2025). Synthetic fibers dominate indoor microplastic profiles, and human activities such as walking, vacuuming, and use of ventilation systems can resuspend particles (Prata, 2018). Additional inhalation sources include occupational settings (e.g., textile, plastic production, or vinyl factories), which may generate high aerosolized plastic dust burdens (Boccia et al, 2024), and emerging sources such as the shedding of fibers from face masks during breathing (Prada et al, 2023). Given the substantial time individuals spend indoors, the respiratory pathway could prove to be the most relevant when considering cumulative microplastic exposure (Cox et al, 2019).

Upon inhalation, the deposition of airborne microplastics within the respiratory tract is strongly influenced by particle size

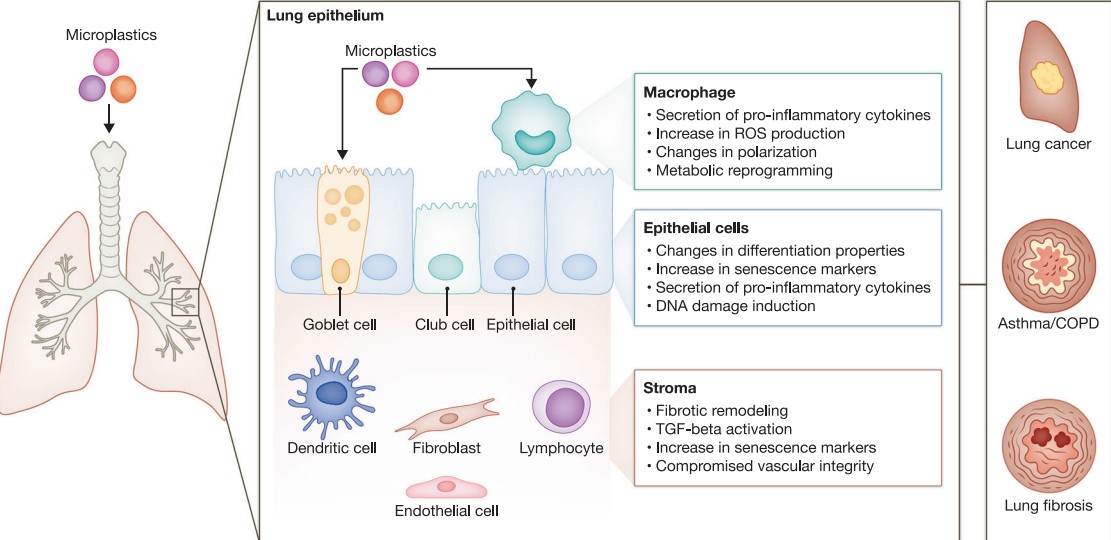

**Figure 2. Overview of the impact of inhaled microplastics on lung cellular homeostasis and associated disease outcomes.**

Following inhalation, microplastics interact with key lung cell populations including macrophages, epithelial cells, and stromal cells, inducing pro-inflammatory cytokine secretion, oxidative stress, DNA damage, cellular senescence, and fibrotic remodeling. Collectively, these cellular disruptions promote the development and exacerbation of severe respiratory conditions such as lung cancer, asthma/COPD, and pulmonary fibrosis.

and aerodynamic properties. Specifically, smaller particles (1–2.5 μm; respirable fraction) readily penetrate into deeper lung compartments, reaching alveolar regions, whereas larger particles (2.5 to 100 μm; inhalable fraction) predominantly deposit within upper conducting airways (Kelly and Fussell, 2012). Furthermore, cellular studies on particles in the nano-scale (i.e., *nanoplastics*) demonstrate either passive penetration through cell membranes or active uptake via endocytosis and/or phagocytosis. They may also, however, remain suspended in air due to their small size, enabling subsequent inhalation. Of note, certain elongated fibers, despite their larger size, could potentially exhibit aerodynamic behaviors that facilitate deep pulmonary deposition (Amato-Lourenço et al, 2020; Bhattacharjee et al, 2024). Consistent with human occupancy patterns and reduced air turnover, indoor environments typically show higher microplastic concentrations compared to outdoor settings, often surpassing 1500 particles per m³ due to prevalent sources such as synthetic textiles, and furnishings (Liao et al, 2021). The most frequently identified airborne polymers include polyethylene terephthalate (PET), polypropylene, and polyethylene, reflecting their extensive application in textiles, packaging, and domestic products (Bhat, 2024). Although methodological limitations currently impede precise quantification, especially for ultrafine particles (<1 μm), recent inhalation exposure estimates indicate daily intake on the order of thousands of particles (Islam et al, 2024; Eberhard et al, 2024). Additionally, detection of microplastics in human lung tissues and bronchoalveolar lavage samples confirms inhalation and deposition within the respiratory tract (Jenner et al, 2022; Qiu et al, 2023; Chen et al, 2022). Thus, despite persistent gaps in epidemiological data, this evidence underscores the ubiquity of airborne microplastic exposure and the need for standardized, high-sensitivity methodologies to quantify human exposure and evaluate potential pulmonary risks.

## How do microplastics affect lung cellular physiology?

The lung represents a dynamic and intricately organized organ, uniquely vulnerable due to its continuous exposure to environmental particulates and pollutants. Upon inhalation, microplastics encounter distinct cellular defenses designed to maintain pulmonary integrity, ranging from frontline immune responders to specialized epithelial barriers and supporting stromal cellular networks. Yet, despite these protective mechanisms, accumulating evidence reveals that microplastics can disrupt lung cellular physiology (Vasse and Melgert, 2024), potentially reshaping respiratory health outcomes and disease trajectories (Fig. 2).

### Macrophages

In the lung, microplastic particles first encounter alveolar macrophages, which reside in the alveolar space and form the lung's phagocytic barrier. These cells internalize particles in a size-dependent manner, with peak phagocytic efficiency for particles around 2–3 μm (Hirota et al, 2007). Studies have suggested that exposure to microplastics in this size range may reduce cell viability and impair macrophage function under certain concentrations and exposure durations (Adler et al, 2024; van den Berg et al, 2025; Ahmadi et al, 2025). However, these effects often occur at higher doses than those likely encountered in environmental inhalation, and dose–response dependencies remain weakly characterized. Moreover, evidence is limited in primary alveolar macrophages or chronic low-dose exposures. Concurrently, studies in macrophage in vitro models and animal systems report that microplastic exposure enhances secretion of pro-inflammatory cytokines such as IL-6, IL-1β, and TNF-α, with limited evidence also suggesting increases in IL-12 or IL-18 under specific experimental conditions

(Yang et al, 2023a; Wolff et al, 2023; Ahmadi et al, 2025). This enhanced cytokine release is proposed to be mechanistically driven by activation of the NLRP3 inflammasome (Yazdi et al, 2010; Sharma et al, 2018), which mediates maturation and secretion of IL-1β and IL-18 (Zaki et al, 2010; Swanson et al, 2019). Furthermore, IL-12 and IL-18 have been suggested to act synergistically to amplify IFN-γ production, intensifying Th1-mediated inflammatory responses (Tominaga et al, 2000).

In addition to the changes in cytokine secretion, a recent study has suggested that exposure to microplastics derived from degraded PET water bottles could lead to increased intracellular reactive oxygen species (ROS) production and a reduction in mitochondrial membrane potential in mouse alveolar macrophages (Tavakol-pournegari et al, 2024). This, interestingly, was accompanied with macrophage polarization towards both M1 (pro-inflammatory) and M2 (anti-inflammatory) phenotypes, with M1 polarization being more pronounced (Tavakolpournegari et al, 2024). Likewise, polystyrene microplastics have been proposed to induce a metabolic shift in macrophages towards glycolysis, accompanied by reduced mitochondrial respiration and upregulation of surface markers CD80 and CD86, consistent with an immunometabolically active state (Merkley et al, 2022). Although most data come from rodent or in vitro models, human macrophages show comparable sensitivity to microplastics exposure, with inter-donor variability in cytokine output (Adler et al, 2024; Weber et al, 2022). Together, these macrophage-driven events create a potent inflammatory environment that could sustain lung inflammation and contribute to long-term respiratory impairment. Yet, the effects of long-term or repeated exposures and responses to environmentally relevant concentrations remain poorly characterized, limiting our understanding of how microplastics impact lung macrophages.

### Respiratory epithelium

The respiratory epithelium forms the second major defense barrier, composed of regionally distinct cell types that include basal, club, goblet, and ciliated cells in the airway, and alveolar type I and II cells in the distal lung. Microplastic exposure appears to affect epithelial cells on multiple levels, as demonstrated in bronchial epithelial lines (BEAS-2B and Calu-3), with recent studies showing that polystyrene and polyethylene particles disrupt cellular metabolism and compromise cell adhesion (Yang et al, 2021; Wu et al, 2024; Lin et al, 2022).

Furthermore, in primary airway epithelial cultures grown at an air–liquid interface, exposure to textile microplastic fibers (e.g., polyester and polyamide/nylon) has been shown to impair epithelial differentiation in a composition- and size-dependent manner, partly due to bioactive leachates released from nylon fibers (Song et al, 2024). These alterations are compounded by the downregulation of tight-junction components, including occludin and claudin-1, which weakens barrier integrity and likely increase epithelial permeability (Yu et al, 2025). Interestingly, inhalable fibrous microplastics, in comparison to the irregular ones, have been observed to disrupt airway epithelial homeostasis more severely, supporting a shape- and composition-dependency of fiber toxicity (Li et al, 2025b). Reviews of microplastic respiratory effects place these findings in the broader context of emerging airway vulnerability to textile microplastics (Li et al, 2025b; Vasse and Melgert, 2024).

Microplastic exposure has been proposed to induce DNA damage in epithelial cells, suggesting a genotoxic potential that could contribute to long-term functional decline (reviewed in detail in (Mahmud et al, 2024)). Consistent with this, microplastics induce cellular senescence, characterized by increased expression of p16 and p21, elevated senescence-associated β-galactosidase activity, and enhanced secretion of pro-inflammatory cytokines that could reinforce chronic inflammation within lung tissues (Jin et al, 2024; Luo et al, 2023). In parallel, ferroptosis (an iron-dependent form of regulated cell death involving lipid peroxidation) has been implicated in microplastic toxicity, as polystyrene microplastics appear to trigger this process in alveolar epithelial cells through activation of the cGAS/STING signaling pathway (Zhang et al, 2024).

Notably, recent work has demonstrated that microplastics not only damage epithelial structure but alter epithelial cell identity. Exposure to nylon microfibers upregulates multiple HOX genes in airway epithelial organoids, including Hoxa4, Hoxa5, Hoxb3, and Hoxc9 (Song et al, 2024). These transcriptional changes, driven by leaching chemicals, were accompanied by reduced differentiation capacity and were partially reversed upon Hoxa5 inhibition, indicating that HOX dysregulation contributes directly to impaired epithelial maturation (Song et al, 2024). Collectively, these findings suggest that synthetic fibers and their leached chemicals can reprogram epithelial lineage commitment and disrupt the regenerative potential of airway epithelia.

### Other lung-resident cell types

Other lung-resident cell types also appear to be affected by microplastic exposure, including fibroblasts and endothelial cells. Fibroblasts, which are central to extracellular matrix maintenance and tissue repair, respond to polystyrene microplastics with increased expression of α-smooth muscle actin (α-SMA) and collagen I, indicative of myofibroblast activation and early fibrotic remodeling (Li et al, 2022). This profibrotic response involves, at least in part, activation of TGF-β signaling, with recent studies implicating transcription factors such as C/EBPβ in promoting fibrotic gene expression programs (Lu et al, 2022; Wang et al, 2022). Pulmonary endothelial cells likewise display microplastic-induced dysfunction, characterized by endothelial activation, upregulation of adhesion molecules and inflammatory cytokines, and potential compromise of vascular integrity, which might lead to barrier disruption and increased monolayer permeability (Vlacil et al, 2021; Lee et al, 2024). These endothelial alterations appear to be aggravated by particle weathering and the co-adsorption of environmental pollutants, suggesting synergistic toxicity relevant to vascular health. Indirectly, immune cell populations, notably dendritic cells and lymphocytes, are likely affected through altered stromal cell signaling and sustained release of inflammatory mediators from fibroblasts and endothelial cells, potentially perpetuating chronic inflammation and impairing immune surveillance.

## Are lung disorders altered by microplastics?

Since microplastics have been shown to functionally affect nearly all major lung cell types, their accumulation could plausibly contribute to the development or progression of pulmonary disease.

Notably, the discovery of microplastic particles in human lung tissue emphasises the gravity of this issue, raising concerns about the long-term health impacts of continued exposure (Jenner et al, 2022; Amato-Lourenço et al, 2021). Occupational studies have already documented adverse respiratory conditions among workers in plastic industries, such as those involved in PVC and nylon production. These studies link high concentrations of inhaled microplastics to serious lung diseases, including lung cancer (Eschenbacher et al, 1999; Atis et al, 2005; Wright and Kelly, 2017).

## Lung cancer

The World Health Organization projects that global cancer cases will surpass 35 million by 2050, representing a 77% increase from the estimated 20 million cases in 2022 (Sung et al, 2021). This trend is particularly concerning for lung cancer, a disease to which most patients succumb within five years of diagnosis. Notably, patterns of lung cancer incidence are evolving alongside environmental change, with recent evidence indicating a continued global increase despite reductions in smoking prevalence (Shankar et al, 2019; Thandra et al, 2021). Increasing evidence indicates that an increasing proportion of lung cancer cases among non-smokers could be driven by ambient air pollution, with fine particulate matter (PM2.5) specifically linked to a heightened risk of lung adenocarcinoma (Shankar et al, 2019; Hill et al, 2023). Alarmingly, a recent study has uncovered new forms of lung cancer in non-smokers that lack canonical *RB1/TP53* alterations and appear to arise through chromothripsis (*massive localized chromosome fragmentation and rearrangements*), further implicating environmental factors in disease development (Sung et al, 2021; Rekhtman et al, 2025).

Due to the abovementioned findings, it is plausible that microplastics contribute to lung cancer initiation and/or progression, particularly as they have been detected more frequently in (pre)malignant lung tissue than in normal tissue (Jenner et al, 2022; Chen et al, 2022; Amato-Lourenço et al, 2021). Inhaled microplastics appear capable of persisting within lung parenchyma, although it remains uncertain whether they play an active role in initiating or amplifying tumor-promoting processes or instead accumulate secondarily in pre-existing lesions. As outlined earlier, microplastics can trigger inflammation and alter epithelial identity in normal lung cells; these effects could converge to create a permissive environment for cancer development and progression. Nevertheless, the cellular and genomic responses of established cancer cells to microplastics remain poorly defined. A limited number of studies in colorectal, glioblastoma, and lung cancer models, however, consistently report that microplastic exposure promotes a migratory phenotype or epithelial-to-mesenchymal transition (EMT) (Traversa et al, 2024; Rafazi et al, 2024; Brynzak-Schreiber et al, 2024). For instance, exposure of human A549 lung cancer cells to polystyrene microplastics significantly reduced cell proliferation but led to formation of filopodia and focal adhesions, features indicative of enhanced motility (Goodman et al, 2021). Comparative analyses further show that polystyrene nanoplastics trigger migration, oxidative stress, and DNA damage more prominently in non-malignant lung cells than in cancer lines, suggesting a role in early tumor-promoting events rather than advanced malignant progression (Ernhofer et al, 2025). However,

exactly how microplastics reprogram cellular and genetic pathways to produce these effects remains unresolved.

Although the effects of microplastics on the epigenomes of lung cancer cells have not yet been investigated, recent findings raise the possibility that lineage-specific gene regulation could be altered upon exposure. Notably, heightened activity of C2H2 zinc-finger transcription factors such as ZNF280C and ZNF865 has been observed as a conserved molecular response to nanomaterials across species (Del Giudice et al, 2023), making them compelling candidates given their established links to cancer phenotypes. Prior work has shown that ZNF280C maintains epigenetic repression at tumor-suppressor loci, preserving H3K27Me3 domains, recruiting SMCHD1, and antagonizing CTCF/cohesin boundary activity, thereby enforcing chromatin compaction and long-range gene silencing (Ying et al, 2022). In colorectal cancer, this function sustains repression of growth-inhibitory programs and correlates with poor prognosis, highlighting an oncogenic role driven by large-scale chromatin reorganization (Ying et al, 2022). By contrast, ZNF865 promotes DNA replication and cell cycle progression through regulation of transcriptional programs that prevent senescence (Levis et al, 2025, 2023). Loss of ZNF865 impairs replication fidelity and induces growth arrest, suggesting that it preserves an epigenomic landscape permissive to proliferation, although its oncogenic potential appears to be context-dependent (Levis et al, 2025, 2023).

While direct evidence for a causal relationship between microplastic exposure and lung cancer development remains elusive, this gap likely reflects both the recent recognition of microplastics as environmental hazards and the longitudinal nature of establishing causality. Nevertheless, the emerging evidence reviewed here supports the hypothesis that microplastics could influence lung cancer biology through epigenetic alterations and modulation of cellular phenotypes critical to tumor initiation and/or progression. With lung cancer incidence projected to rise alongside increasing environmental pollution, there is a growing need for mechanistic studies that delineate how microplastic exposure affects lung tissue biology. Such work could identify early molecular changes and pathways amenable to preventive or therapeutic intervention.

## Obstructive lung disorders

Similar to cancer, the incidence of obstructive lung disorders, particularly asthma and chronic obstructive pulmonary disease (COPD), continues to rise globally. Curiously, over the past several decades, both disorders have demonstrated shifts in clinical presentations, becoming increasingly severe and resistant to conventional therapies (Bell and Busse, 2013; Chapman and McIvor, 2010; Bollmeier and Hartmann, 2020; Rosenwasser et al, 2022). Asthma phenotypes have diversified, with a growing prevalence of non-atopic forms characterized by neutrophilic inflammation, reduced corticosteroid responsiveness, and heightened symptom severity (Kuruvilla et al, 2019; Xie et al, 2022; Wadhwa et al, 2019; Liu et al, 2024). Similarly, COPD is now increasingly recognized to present with earlier onset and a more aggressive course, featuring accelerated loss of lung function and frequent exacerbations associated with chronic inflammation and epithelial barrier dysfunction (Rehman et al, 2021; Beijers et al, 2022; Liu et al, 2023). These evolving disease trajectories closely

parallel increasing exposure to environmental pollutants, implicating airborne contaminants, including microplastics, as potential contributors to these emerging clinical profiles (Gou et al, 2024; Paplińska-Goryca et al, 2025; Aghapour et al, 2022; Cheng et al, 2025).

Consistent with epidemiological observations, murine experimental models have provided evidence that exposure to polystyrene microplastics worsens both asthma and COPD phenotypes, primarily by promoting inflammation. In asthma models, microplastics trigger profound airway inflammation accompanied by eosinophilic infiltration, mucus overproduction, and heightened airway reactivity, leading to distinct histopathological alterations (Wang et al, 2025; Wei et al, 2024). In COPD models, exposure similarly amplifies airway inflammation, characterized by neutrophil-dominated responses, oxidative stress, and alveolar destruction, together with increased susceptibility to ferroptosis and autophagy-related cell injury (Yang et al, 2024; Wei et al, 2025).

Mechanistically, microplastics have been proposed to exacerbate asthma predominantly through augmentation of Th2-mediated inflammation, characterized by elevated expression of cytokines such as IL-4, IL-5, and IL-13 (Wei et al, 2024; Wang et al, 2025; Xu et al, 2025). A proposed mechanism underlying this response is the microplastic-induced secretion of extracellular Heat Shock Protein 90α (HSP90α), a molecular chaperone released during cellular stress that contributes to immune activation and epithelial dysfunction. Notably, pharmacological inhibition of HSP90α significantly attenuated microplastic-induced airway inflammation, identifying it as both a potential biomarker and a promising therapeutic target (Xu et al, 2025). In addition, combined exposure to polystyrene microplastics and di-(2-ethylhexyl) phthalate (DEHP), a commonly used plasticizer, further intensifies allergic airway inflammation, increasing Th2 cytokine expression, eosinophil infiltration, and oxidative stress. These synergistic effects are mediated primarily through activation of the TRPA1-p38 MAPK signaling cascade, highlighting the complex interactions among environmental pollutants that drive severe asthma phenotypes (Han et al, 2023). Transcriptional analyses further reveal that polyamide microplastics reprogram epithelial cells from asthma patients, upregulating genes involved in sterol and cholesterol biosynthesis, secondary alcohol metabolism, and acetyl-CoA pathways - changes consistent with metabolic remodeling that may further perpetuate inflammation (Han et al, 2023; Paplińska-Goryca et al, 2025).

In the context of COPD, emerging evidence positions polystyrene microplastics as potential exacerbators of disease progression through the induction of ferroptosis driven by autophagy-dependent mitochondrial dysfunction (Wei et al, 2025; Liu and Chen, 2017). Mechanistically, microplastic accumulation in lung tissue provokes mitochondrial dysfunction, with excessive ROS production driving lysosomal activation and metabolic stress within epithelial cells (Wei et al, 2025). These impairments subsequently trigger ferritinophagy, leading to iron overload and lipid peroxidation, hallmark events of ferroptotic cell death, which together amplify pulmonary inflammation and worsen COPD pathology. Consistent with these findings, pharmacological inhibition of mitochondrial ROS or ferroptosis effectively alleviates acute microplastic-induced COPD exacerbations, highlighting potential therapeutic avenues. Complementary transcriptional profiling reveals that microplastic exposure reprograms epithelial cells from COPD patients toward a pro-inflammatory state, marked by enhanced chemokine activity and impaired barrier function (Paplińska-Goryca et al, 2025). Markedly, both asthma and COPD display elevated expression of IL-19 and BCL2L15 genes upon microplastic exposure, linking these pollutants to a shared axis of Th2-associated inflammation and potentially carcinogenic signaling (Paplińska-Goryca et al, 2025).

Taken together, accumulating epidemiological and experimental evidence underscores a connection between microplastic exposure and the worsening of obstructive lung disorders such as asthma and COPD. Microplastics appear to promote more severe, therapy-resistant disease phenotypes through overlapping inflammatory and metabolic mechanisms, including Th2-driven cytokine release, mitochondrial dysfunction, and ferroptosis induction. Given these insights, targeted investigation into microplastic-driven molecular pathways is warranted, as such work may help elucidate mechanisms and markers of pollutant-induced respiratory decline.

## Lung fibrosis

Pulmonary fibrosis, historically characterized as a chronic disease driven primarily by aging and genetic predispositions, has shown notable shifts in clinical presentation and severity over recent decades (Kolb and Collard, 2014; Wolters et al, 2018). Beyond the rising incidence rates, clinical observations reveal diversification of pulmonary fibrosis phenotypes, including aggressive forms such as rapidly progressive idiopathic pulmonary fibrosis and the combined fibrosis–emphysema form (Kaul et al, 2022; Olson et al, 2018; Soares Pires et al, 2013; Papaioannou et al, 2016). These emerging phenotypes frequently present with early respiratory failure, rapid functional decline, and distinctive radiographic features, often unresponsive to conventional anti-fibrotic treatments. Genetic studies have helped clarify some of the underlying mechanisms of these shifts, implicating variants such as the *MUC5B* promoter polymorphism as a major modifier of disease susceptibility and progression (Seibold et al, 2011; Schwartz et al, 2022). However, genetic factors alone cannot fully explain this diversification. Increasing evidence points to a substantial role for environmental pollutants, particularly inhaled particulates, in reshaping disease trajectories. Epidemiological studies consistently link exposures such as smoking, ambient air pollution, and occupational inhalants with greater disease severity and incidence, highlighting an expanding environmental contribution to pulmonary fibrosis (Sack et al, 2025; Park et al, 2024).

Emerging experimental data implicate inhaled polystyrene microplastics in the initiation and progression of pulmonary fibrosis (Li et al, 2022; Yang et al, 2024). In murine models, long-term intranasal or intratracheal exposure induces fibrotic remodeling characterized by increased collagen deposition and elevated expression of fibrotic markers such as α-SMA, vimentin, and Col1a, alongside epithelial injury, alveolar wall thickening, and inflammatory infiltration (Li et al, 2022; Wei et al, 2024).

Oxidative stress appears central to this process, with reduced antioxidant enzyme activity (SOD and GSH-Px) driving activation of profibrotic pathways, notably Wnt/β-catenin (Li et al, 2022; Wei et al, 2024; Chilosi et al, 2003). Recent work extends these findings by demonstrating that polystyrene microplastics also induce ferroptosis through activation of the cGAS/STING pathway,

leading to iron accumulation, lipid peroxidation, and loss of cellular redox balance (Zhang et al, 2024). Pharmacologic inhibition of ferroptosis (e.g., Fer-1) or blockade of cGAS/STING signaling (e.g., G150/H151) markedly attenuates fibrosis, underscoring the therapeutic relevance of these pathways (Zhang et al, 2024) Complementary studies using polystyrene nanoplastics in bronchial epithelial models demonstrate ferroptotic-like cell death mediated by the HIF-1α/HO-1 axis, suggesting that both micro- and nano-sized particles can activate convergent redox and fibrogenic pathways (Zhang et al, 2024; Wang et al, 2025). Moreover, PVC microplastics have been demonstrated to induce epithelial senescence via ROS signaling in lung models, a finding consistent with a shared oxidative stress axis in microplastic-driven pathology (Jin et al, 2024).

Additional mechanistic insights point to broader disruptions of lung cellular and immune homeostasis. Prolonged microplastic exposure compromises epithelial barrier integrity, facilitating deeper penetration of pollutants and pathogens. Altered pulmonary microbiota composition, marked by enrichment of Gram-negative bacteria and elevated lipopolysaccharide (LPS) release, can activate Toll-like receptor 4 (TLR4) signaling, disturb iron metabolism, and trigger ferroptotic injury (Kang et al, 2024).

In parallel, microplastics also promote EMT-switch, with downregulation of epithelial markers such as E-cadherin and induction of mesenchymal proteins that contribute to tissue scarring (Traversa et al, 2024). Persistent activation of inflammatory pathways, particularly via NF-κB and NLRP3 inflammasome activation, further amplifies injury and promotes collagen deposition (Alijagic et al, 2023).

In summary, emerging evidence suggests that inhaled microplastics contribute to the pathogenesis of pulmonary fibrosis. Through above-discussed interrelated mechanisms, microplastics appear capable of disrupting lung tissue homeostasis and promoting persistent fibrotic remodeling. As evidence accumulates on the rising incidence and heterogeneity of pulmonary fibrosis, ongoing mechanistic and translational research will be important to improve understanding of disease mechanisms and explore potential avenues for intervention.

# Discussion

Historically, key developments in public health have hinged on our capacity to identify and address environmental threats in a timely manner. The widespread presence of microplastics in modern environments highlights the importance of maintaining such awareness today. Far from being inert fragments of industrial convenience, microplastics constitute a complex, persistent, and biologically active form of particulate pollution. The distinct physicochemical characteristics of microplastics allow them to interact with pulmonary tissues, potentially eliciting inflammatory signaling, perturbing cellular homeostasis, and modifying cell fate, thereby contributing to respiratory disease processes.

What is notable is not merely the breadth of cellular disruption caused by microplastics, but their potential to influence disease-related pathways. Emerging evidence indicates that microplastics may modulate molecular and inflammatory processes relevant to lung cancer, COPD, asthma, and fibrosis. Experimental studies further suggest that microplastics can induce EMT-like changes,

alter cellular metabolism, and affect genomic or epigenomic regulation, raising concern that they could influence disease development and progression. Collectively, these findings call for a reconsideration of how environmental pollutants shape human health, revealing a previously underappreciated dimension of pulmonary vulnerability.

Yet, recognizing microplastics as potential contributors to disease progression offers an important opportunity for advancing environmental health research. The growing body of evidence highlights the need for coordinated scientific innovation and evidence-based regulatory consideration. By addressing this challenge proactively rather than reactively, the scientific community can strengthen understanding of microplastic toxicity and inform preventive strategies. Moving beyond documentation toward mechanistic insight will be key to guiding effective interventions and ensuring that emerging environmental health risks are managed with foresight rather than hindsight.

While establishing biologically meaningful thresholds for microplastic exposure would mark a crucial step forward, the data required to define such relationships remain fragmentary and inconsistent. Reported concentrations of airborne microplastics differ by several orders of magnitude, from fewer than ten to more than 1500 particles per cubic meter depending on sampling strategy, setting, and analytical approach (O'Brien et al, 2023; Zhu et al, 2021; Yakovenko et al, 2025). Modeled inhalation loads are equally variable, ranging from a few dozen to tens of thousands of particles per day when indoor air and fiber-rich environments are considered (Vianello et al, 2019; Yakovenko et al, 2025; Chen et al, 2025; Wardani et al, 2024). Yet, these estimates largely omit the submicron fraction most capable of reaching the alveolar surface, leaving true pulmonary exposure unresolved. In contrast, experimental systems typically rely on exposure levels vastly exceeding plausible environmental concentrations (*often tens to hundreds of micrograms per milliliter*) selected to provoke measurable oxidative and inflammatory responses. This disparity illustrates the challenge of reconciling environmentally relevant exposures with experimental findings, thereby limiting current attempts to define robust dose–response thresholds. Comparative studies systematically examining how particle size, shape, and polymer chemistry modulate toxicity remain exceedingly rare. In light of this diversity, it is notable that most existing lung studies still employ spherical particles (Table 1), offering only a limited view of the biological effects likely to arise from the more irregular and fibrous forms that dominate real airborne exposure. Bridging these gaps will require coordinated efforts to establish standardized exposure-response frameworks that integrate environmental monitoring, toxicokinetic modeling, and mechanistic experimentation —foundations upon which any credible risk assessment must ultimately rest.

To effectively confront this emerging threat, the scientific community must urgently quantify real-world human exposure, particularly among vulnerable groups such as children, the elderly, and those with pre-existing conditions, while investing in highly sensitive methodologies capable of detecting ultrafine plastic particles in biological tissues. Parallel epidemiological and mechanistic research is needed to establish clear exposure-disease relationships, clarify the long-term consequences of chronic low-dose inhalation, and identify biomarkers that enable early detection and intervention. At the same time, proactive public health measures

(i.e., stricter regulation of synthetic microplastic production, improved indoor air quality standards, and broad educational campaigns) are essential to reduce exposure at the population level. Only through coordinated efforts across science, medicine, policy, and public engagement can we begin to address and mitigate the potential respiratory health burden posed by microplastics.

## Pending issues

- True pulmonary exposure to microplastics remains undefined; submicron fractions are largely unquantified.
- Comparative toxicology across polymer types, shapes, and aged versus pristine particles is missing.
- Mechanistic integration of oxidative stress, ferroptosis, and cell differentiation is incomplete.
- Causal contribution of microplastics to lung cancer, COPD, asthma, and fibrosis remains unresolved.
- Synergistic toxicity with co-pollutants such as PM 2.5, volatile organic compounds, and plasticizers is poorly characterized.
- Vulnerable groups (children, elderly, patients with pre-existing lung disease or genetic susceptibility) are underrepresented.
- Lack of chronic low-dose inhalation models and validated biomarkers limits translation to human disease.
- Standardized exposure–response frameworks are urgently needed to support risk assessment and regulation.

## Peer review information

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

## Acknowledgements

This work was funded by the Netherlands Organization for Scientific Research NWO OpenCompetition grant 09120242410075. The authors would like to thank Sarah Vahed for proofreading and giving suggestions on the text. Additionally, members of both Prekovic and Melgert labs for their comments on the manuscript.

## Author contributions

**Emmanouela Epeslidou**: Conceptualization; Visualization; Writing—original draft; Writing—review and editing. **Julia S Scott**: Writing—original draft; Writing—review and editing. **Bim de Klein**: Writing—review and editing. **Jeremy Tan Cudia**: Writing—review and editing. **Barbro Melgert**: Conceptualization; Supervision; Funding acquisition; Writing—original draft; Writing—review and editing. **Stefan Prekovic**: Conceptualization; Supervision; Funding acquisition; Visualization; Writing—original draft; Writing—review and editing.

## Disclosure and competing interests statement

The authors declare no competing interests.

