## [Peer Review File · EMBO Molecular Medicine]

Microplastics as environmental modifiers of lung disease

Emmanouela Epeslidou, Julia Scott, Bim de Klein, Jeremy Cudia, Barbro Melgert, and Stefan Prekovic

Corresponding authors: Stefan Prekovic (s.prekovic@umcutrecht.nl) , Barbro Melgert (b.n.melgert@rug.nl)

Review Timeline:

Submission Date:	15th Jul 25
Editorial Decision:	4th Aug 25
Revision Received:	8th Oct 25
Editorial Decision:	30th Oct 25
Revision Received:	11th Nov 25
Accepted:	20th Nov 25

Editor: Lise Roth

Transaction Report:

4th Aug 2025

Dear Stefan,

Thank you for the submission of your review to EMBO Molecular Medicine. We have now received feedback from the experts who agreed to evaluate your manuscript. As you will see from the reports below, they overall found the review well written, new, and interesting. They nevertheless raise a few concerns and make several suggestions to improve the interest and impact of your work.

We would therefore welcome a revised version of your manuscript that would address these points. Please attach a covering letter giving details of the way in which you have handled each of the points raised by the referees.

- 1/ A .doc formatted version of the manuscript text (including Figure legends and tables).
- 2/ Separate figure files.
- 3/ A letter INCLUDING the reviewer's reports and your detailed responses to their comments.
- 4/ A glossary: EMBO Molecular Medicine articles are accompanied by a glossary explaining some of the terms used for laymen.
- 5/ Pending issues: At the end of each article, there is a box highlighting issues that still need further studies and where research efforts should converge.
- 6/ A 'disclosure statement and competing interests' statement (<https://www.embopress.org/competing-interests>).
- 7/ Up to 5 keywords.
- 8/ Please remove the Authors Contributions from the manuscript and use the free text boxes beneath each contributing author's name in our system to add specific details on the author's contribution. More information is available in our guide to authors.
- 9/ References should be in alphabetical order, with 10 authors before et al.

For the figures, please note the following points:

- If there are certain aspects of your figure draft that are based upon assumptions or where the scientific data remains ambiguous, please add a comment so that we can work with you on an accurate depiction.
- If the figure or single panels of the figure have been adapted from a published figure, please add this information to the figure legend (e.g., 'Adapted from...' or 'Based on...').
- Please only re-use figures or parts of a figure if this is essential for understanding the concept communicated. If the figure contains re-used images or elements of images, please make sure that you have the permission/license to publish it (this also applies to your own previous work, if the journal you published in retains copyright). All re-used material must be explicitly cited.
- If you use an image data base for scientific iconography (e.g., BioRender), please let us know if you have a license that allows for publication in an academic journal.

Looking forward to receiving your revised manuscript,

With kind regards,

Lise

***** Reviewer's comments *****

Referee #1 (Remarks for Author):

This review addresses a timely and important topic—the health impacts of microplastic exposure, particularly in relation to pulmonary diseases. The authors have compiled a substantial body of evidence on cellular responses to microplastics and their potential contributions to respiratory pathologies. However, the manuscript would benefit from a more cohesive and mechanistic synthesis of how specific physicochemical characteristics of microplastics influence biological outcomes. Clarifying disease-specific hypotheses, identifying relevant exposure thresholds, and outlining concrete research and clinical priorities would further enhance the review's impact. Overall, the manuscript is clearly written and accessible. The following major and minor concerns are offered to help strengthen the manuscript.

Major Concerns

The review highlights several cellular responses elicited by microplastics—such as inflammation, oxidative stress, and ferroptosis—but lacks sufficient mechanistic insight into how distinct physicochemical characteristics of microplastics (e.g., size, shape, polymer composition, and degree of environmental aging) differentially contribute to these outcomes. For example, the biological effects of fibrous microplastics compared to fragmented particles, or differences between polymers like polyethylene and PVC, are only superficially addressed. The influence of aging-related surface modifications or interactions with co-contaminants (e.g., heavy metals, plasticizers) is acknowledged but not thoroughly integrated into the mechanistic narrative. To improve coherence, the authors should consider dedicating a focused section discussing how traits such as particle size (<2.5 µm), surface charge, and polymer type influence specific signaling pathways, such as NLRP3 inflammasome activation or cGAS/STING signaling, and how these interactions may drive distinct disease phenotypes. Comparative data—such as toxicity profiles across polymer types and pristine vs. aged particles—would strengthen this aspect of the review.

The section on lung diseases catalogs various reported associations between microplastic exposure and pulmonary outcomes but does not adequately synthesize overarching mechanisms or highlight unresolved controversies. For instance, while the detection of microplastics in lung tumors is noted, it remains unclear whether these particles contribute causally to tumorigenesis or merely accumulate passively due to altered tissue permeability. Similarly, the reported role of microplastics in therapy resistance in obstructive lung disease is not contextualized in relation to known environmental cofactors like PM2.5. Reframing the disease sections (e.g., lung cancer, asthma/COPD, fibrosis) around a few central questions—such as whether microplastics independently promote disease progression, or how they interact with host immune or epithelial responses—would provide greater analytical depth. It is also worth critically appraising contradictory findings, such as inconsistent reports on macrophage polarization or fibroblast activation, to underscore knowledge gaps.

While the review notes the difficulty in quantifying microplastic exposure, it stops short of connecting reported concentrations to plausible biological thresholds for disease induction. For example, ambient exposure levels are mentioned, but it remains unclear whether these concentrations reach the levels necessary to trigger measurable biological responses—such as inflammasome activation or epithelial barrier disruption—in vivo. Moreover, vulnerable populations, such as children, the elderly, or individuals with pre-existing lung disease, are only briefly referenced and not meaningfully integrated into the core analysis. The review would benefit from a dedicated section on "Exposure Metrics and Risk Stratification," integrating available data on microplastic concentrations in indoor air, dust, and human tissues with known cellular thresholds for dysfunction. Additionally, the review should consider host factors—such as comorbidities, smoking status, or genetic predispositions (e.g., MUC5B variants in pulmonary fibrosis)—that may modulate susceptibility to microplastic-related damage.

The "Call to Action" section outlines general priorities but lacks specificity in guiding research, regulatory, or clinical responses. For example, while the need for improved detection technologies is emphasized, the review does not specify which techniques (e.g., Raman vs. µFTIR) are most suitable for identifying nanoplastics in biological tissues. Similarly, potential biomarkers of exposure or injury are mentioned without addressing their clinical utility or validation status. This section should be restructured into a more actionable roadmap, covering: (i) priority research directions (e.g., chronic low-dose inhalation studies in relevant models), (ii) methodological gaps (e.g., harmonized protocols for nanoplastic quantification), (iii) policy recommendations (e.g., restrictions on microplastic additives in textiles), and (iv) clinical implications (e.g., candidate biomarkers like HSP90α or cGAS/STING for disease monitoring).

Minor Concerns

Some sections of the review are imbalanced in coverage and depth. For instance, the discussion of microplastic physicochemical properties is detailed, yet the sections on ingestion and dermal exposure are relatively underdeveloped. Given the evidence suggesting that microplastics absorbed via the gastrointestinal tract may reach the lungs through systemic circulation, the review should expand on these alternative exposure routes. Additionally, the lung fibrosis section is more comprehensive than the lung cancer section, which would benefit from a deeper exploration of epigenetic mechanisms—such as roles for ZNF280C or chromatin remodeling—in mediating microplastic-related oncogenic changes.

In several instances, the manuscript reiterates similar findings from different studies (e.g., polystyrene-induced ROS in macrophages), which could be condensed. A tabulated summary categorizing cellular responses (e.g., oxidative stress, apoptosis, fibrosis) by particle type, size, and polymer class would help readers identify consistent trends and variability. This would also allow the text to focus more on interpretation rather than repetition. For example, the authors could highlight that fibrous particles, regardless of polymer type, consistently elicit stronger fibrogenic responses than spherical particles, or that aged microplastics appear to have greater immunotoxicity than pristine ones.

Referee #2 (Remarks for Author):

The topic is timely and important. The summary of mechanistic research is helpful. However, there are key elements that need to be corrected before publication:

Misleading - The link between mechanistic changes and disease needs to be carefully and accurately described. Evidence isn't there yet for some of the statements - e.g. authors made leap from mechanistic changes to "profound influence on disease trajectories":

What is especially striking is not merely the breadth of cellular disruption caused by microplastics, but their profound influence on disease trajectories themselves. Specifically, microplastics are emerging as key active modifiers that drive the aggressive, therapy-resistant phenotypes increasingly seen in lung cancer, COPD, asthma, and fibrosis.

-----Focus on inhalation sources needed: Only gave 2 sentences on inhalation exposure sources. e.g wildfires, occupational exposure sources. Need much more on inhalation exposure sources this if paper is proposing public health changes, etc...

-----Need clarity on research type: Needs to better distinguish when discussing between human vs. animal studies, and direct findings vs. surmised implications. An uneducated reader, it would be very difficult to distinguish as it's written.

-----Avoiding exaggerated/extreme language: Maintain an evidence-based tone throughout. Remove all extreme language present throughout the text.

-----Substantiating claims: Ensuring statements are supported or qualified (i.e., lack of references): e.g. the following needs references - "Studies have suggested that exposure to microplastics in this range reduces macrophage viability, impairs uptake, and induces oxidative stress..."

Referee #3 (Remarks for Author):

In this manuscript the authors extensively review of microplastics, persistent synthetic polymer particles, have emerged as an environmental hazard, notably affecting pulmonary health due to continuous inhalation exposure. The authors try to link the airborne microplastics and severe lung diseases, including cancer, asthma, COPD, and lung fibrosis, and provide the potential underlying mechanistic pathways. The review integrates emerging epidemiological evidence with recent mechanistic insights, highlighting how inhaled microplastics disrupt lung cellular homeostasis by inducing inflammation, oxidative stress, ferroptosis, epithelial-mesenchymal transition, and epigenetic alterations.

This review is very timely and important. It is very comprehensive. My suggestions:

1. The manuscript is too long, more than 5,000 word counts and 132 references. It's may be more friendly to readers if the manuscript can be reduced to at least one-third or half.
2. A table to summarize the clinical relevance of each individual lung disease and microplastics exposure, the current epidemiologic evidence, mechanisms and related references. It will be even more readable.
3. In the introduction, "Recent studies demonstrate the systemic presence of microplastics, with particles detected in human blood, lung, heart, testes, and brain [3-7], raising significant concerns about bioaccumulation and disease risk." Presence of microplastics is not necessarily related to the diseases. Please extend.
4. In page 9, Are lung disorders altered by microplastics?
"Occupational studies have already documented adverse respiratory conditions among workers in plastic industries, such as those involved in PVC and nylon production. These studies link high concentrations of inhaled microplastics to serious lung diseases, including lung cancer." Please cite reference for this sentence.

Rebuttal letter*Epeslidou et al.*

We sincerely thank all reviewers for their constructive and thoughtful feedback. We deeply appreciate the time and expertise invested in evaluating our manuscript. The reviewers' comments have been instrumental in refining the clarity, precision, and overall focus of our review. We have carefully considered each point and implemented substantial revisions accordingly. Below, we provide a detailed, point-by-point response outlining how we have addressed each suggestion and improved the manuscript in line with the reviewers' recommendations.

Referee #1

This review addresses a timely and important topic—the health impacts of microplastic exposure, particularly in relation to pulmonary diseases. The authors have compiled a substantial body of evidence on cellular responses to microplastics and their potential contributions to respiratory pathologies. However, the manuscript would benefit from a more cohesive and mechanistic synthesis of how specific physicochemical characteristics of microplastics influence biological outcomes. Clarifying disease-specific hypotheses, identifying relevant exposure thresholds, and outlining concrete research and clinical priorities would further enhance the review's impact. Overall, the manuscript is clearly written and accessible. The following major and minor concerns are offered to help strengthen the manuscript.

Major Concerns

The review highlights several cellular responses elicited by microplastics—such as inflammation, oxidative stress, and ferroptosis—but lacks sufficient mechanistic insight into how distinct physicochemical characteristics of microplastics (e.g., size, shape, polymer composition, and degree of environmental aging) differentially contribute to these outcomes. For example, the biological effects of fibrous microplastics compared to fragmented particles, or differences between polymers like polyethylene and PVC, are only superficially addressed. The influence of aging-related surface modifications or interactions with co-contaminants (e.g., heavy metals, plasticizers) is acknowledged but not thoroughly integrated into the mechanistic narrative. To improve coherence, the authors should consider dedicating a focused section discussing how traits such as particle size (<2.5 μm), surface charge, and polymer type influence specific signaling pathways, such as NLRP3 inflammasome activation or cGAS/STING signaling, and how these interactions may drive distinct disease phenotypes. Comparative data—such as toxicity profiles across polymer types and pristine vs. aged particles—would strengthen this aspect of the review.

We thank the reviewer for this insightful comment. In response, we have added a dedicated paragraph discussing how distinct physicochemical characteristics of microplastics (e.g. *polymer type, particle size, shape, and degree of environmental aging*) differentially influence biological outcomes. In line with Reviewer #3's suggestion, we have also introduced a new summary table (*Table 1*) that integrates these relationships, outlining polymer class, particle dimensions, and morphology alongside the associated

cellular responses and biological effects. The new paragraph appears on page 5, and the corresponding table is provided at the end of the revised manuscript (pages 33-36).

The section on lung diseases catalogs various reported associations between microplastic exposure and pulmonary outcomes but does not adequately synthesize overarching mechanisms or highlight unresolved controversies. For instance, while the detection of microplastics in lung tumors is noted, it remains unclear whether these particles contribute causally to tumorigenesis or merely accumulate passively due to altered tissue permeability. Similarly, the reported role of microplastics in therapy resistance in obstructive lung disease is not contextualized in relation to known environmental cofactors like PM_{2.5}. Reframing the disease sections (e.g., lung cancer, asthma/COPD, fibrosis) around a few central questions—such as whether microplastics independently promote disease progression, or how they interact with host immune or epithelial responses—would provide greater analytical depth. It is also worth critically appraising contradictory findings, such as inconsistent reports on macrophage polarization or fibroblast activation, to underscore knowledge gaps.

We thank the reviewer for this valuable and constructive comment. In response, we have substantially revised all disease-related sections to provide a more cohesive mechanistic synthesis and to better address unresolved controversies. Each section is now framed around central mechanistic questions. We have clarified the uncertainty regarding causality in lung cancer (*i.e.*, *passive accumulation versus active initiation*), contextualized the discussion of obstructive lung diseases in relation to established environmental cofactors such as PM_{2.5}, and highlighted areas where findings remain inconsistent or limited, particularly concerning macrophage polarization and fibroblast activation.

While the review notes the difficulty in quantifying microplastic exposure, it stops short of connecting reported concentrations to plausible biological thresholds for disease induction. For example, ambient exposure levels are mentioned, but it remains unclear whether these concentrations reach the levels necessary to trigger measurable biological responses—such as inflammasome activation or epithelial barrier disruption—in vivo. Moreover, vulnerable populations, such as children, the elderly, or individuals with pre-existing lung disease, are only briefly referenced and not meaningfully integrated into the core analysis. The review would benefit from a dedicated section on "Exposure Metrics and Risk Stratification," integrating available data on microplastic concentrations in indoor air, dust, and human tissues with known cellular thresholds for dysfunction. Additionally, the review should consider host factors—such as comorbidities, smoking status, or genetic predispositions (e.g., MUC5B variants in pulmonary fibrosis)—that may modulate susceptibility to microplastic-related damage.

In the revised manuscript, we have expanded the discussion of exposure quantification to clarify both methodological and conceptual limitations. We now specify that reported airborne microplastic concentrations range from fewer than ten to over 1,500 particles per m³, depending on sampling strategy

and analytical approach, while modeled inhalation loads vary from a few dozen to tens of thousands of particles per day, particularly in fiber-rich indoor environments. We further emphasize that these estimates largely omit the submicron fraction most relevant for alveolar deposition, and that most experimental studies still rely on exposure levels several orders of magnitude higher than those plausibly encountered in the environment - currently precluding definition of biologically meaningful thresholds for disease induction. Finally, we have expanded the discussion of population vulnerability, highlighting how factors such as age, pre-existing lung disease, and genetic susceptibility (e.g., MUC5B variants) may modulate individual risk once more accurate exposure–response data become available.

The "Call to Action" section outlines general priorities but lacks specificity in guiding research, regulatory, or clinical responses. For example, while the need for improved detection technologies is emphasized, the review does not specify which techniques (e.g., Raman vs. μ FTIR) are most suitable for identifying nanoplastics in biological tissues. Similarly, potential biomarkers of exposure or injury are mentioned without addressing their clinical utility or validation status. This section should be restructured into a more actionable roadmap, covering: (i) priority research directions (e.g., chronic low-dose inhalation studies in relevant models), (ii) methodological gaps (e.g., harmonized protocols for nanoplastic quantification), (iii) policy recommendations (e.g., restrictions on microplastic additives in textiles), and (iv) clinical implications (e.g., candidate biomarkers like HSP90 α or cGAS/STING for disease monitoring).

We agree that the Call to Action should offer clearer direction for future research and regulatory priorities. In line with Reviewer #3's recommendation to streamline the manuscript, we merged the concluding sections into a single, integrated Discussion / Call to Action. The revised section now highlights key priorities, including the development of standardized exposure–response frameworks, improved detection methodologies for nanoplastics, quantification of real-world exposure with attention to vulnerable populations, and stronger coordination between scientific, clinical, and policy domains.

Minor Concerns

Some sections of the review are imbalanced in coverage and depth. For instance, the discussion of microplastic physicochemical properties is detailed, yet the sections on ingestion and dermal exposure are relatively underdeveloped. Given the evidence suggesting that microplastics absorbed via the gastrointestinal tract may reach the lungs through systemic circulation, the review should expand on these alternative exposure routes. Additionally, the lung fibrosis section is more comprehensive than the lung cancer section, which would benefit from a deeper exploration of epigenetic mechanisms-such as roles for ZNF280C or chromatin remodeling-in mediating microplastic-related oncogenic changes.

We thank the reviewer for this helpful comment. We have made efforts to improve the overall balance of the manuscript by expanding sections where possible, while recognizing that the current literature

remains uneven across exposure routes and disease models. In particular, we have clarified the limited evidence available on ingestion and systemic exposure, and strengthened mechanistic discussion where data permit. The remaining imbalance in depth largely reflects the scarcity of studies on certain pathways, particularly systemic translocation and epigenetic mechanisms in lung cancer.

In several instances, the manuscript reiterates similar findings from different studies (e.g., polystyrene-induced ROS in macrophages), which could be condensed. A tabulated summary categorizing cellular responses (e.g., oxidative stress, apoptosis, fibrosis) by particle type, size, and polymer class would help readers identify consistent trends and variability. This would also allow the text to focus more on interpretation rather than repetition. For example, the authors could highlight that fibrous particles, regardless of polymer type, consistently elicit stronger fibrogenic responses than spherical particles, or that aged microplastics appear to have greater immunotoxicity than pristine ones.

We have condensed repetitive descriptions of similar findings, particularly in sections discussing oxidative stress and inflammatory responses. In addition, we have incorporated a summary table categorizing key cellular responses (e.g., oxidative stress, apoptosis, fibrosis) by particle type, size, and polymer class to help readers identify consistent trends and variability across studies. We also highlight in the text that fibrous and environmentally aged particles generally elicit stronger fibrogenic and immunotoxic responses than spherical or pristine particles, reflecting emerging consensus in the literature.

Referee #2

The topic is timely and important. The summary of mechanistic research is helpful. However, there are key elements that need to be corrected before publication:

Misleading - The link between mechanistic changes and disease needs to be carefully and accurately described. Evidence isn't there yet for some of the statements - e.g. authors made leap from mechanistic changes to "profound influence on disease trajectories": What is especially striking is not merely the breadth of cellular disruption caused by microplastics, but their profound influence on disease trajectories themselves. Specifically, microplastics are emerging as key active modifiers that drive the aggressive, therapy-resistant phenotypes increasingly seen in lung cancer, COPD, asthma, and fibrosis.

We thank the reviewer for this valuable comment. We agree that the evidence linking cellular and molecular alterations to clinical disease trajectories remains preliminary and should be interpreted with caution. In response, we have carefully revised the phrasing throughout the manuscript, particularly in the specified section, to avoid implying a causal relationship. The text now states that microplastics *may* influence rather than *profoundly alter* disease trajectories, reflecting their emerging yet unconfirmed role as potential modulators of disease progression and severity.

Focus on inhalation sources needed: Only gave 2 sentences on inhalation exposure sources. e.g wildfires, occupational exposure sources. Need much more on inhalation exposure sources this if paper is proposing public health changes, etc...

We thank the reviewer for this valuable comment. In response, we have expanded the section on inhalation exposure to provide a more comprehensive overview of key sources (see page 6).

Need clarity on research type: Needs to better distinguish when discussing between human vs. animal studies, and direct findings vs. surmised implications. An uneducated reader, it would be very difficult to distinguish as it's written.

We thank the reviewer for this valuable comment. In response, we have clarified throughout the text whether findings originate from human, animal, or *in vitro* studies and have been more explicit in distinguishing direct evidence from inferred implications. Additionally, we have included this information in the new summary table to further improve clarity and accessibility for readers (see Table 1; pages 33-36).

Avoiding exaggerated/extreme language: Maintain an evidence-based tone throughout. Remove all extreme language present throughout the text.

We thank the reviewer for this comment. We have revised the manuscript to remove exaggerated language and ensure an evidence-based, balanced tone throughout.

Substantiating claims: Ensuring statements are supported or qualified (i.e., lack of references): e.g. the following needs references - "Studies have suggested that exposure to microplastics in this range reduces macrophage viability, impairs uptake, and induces oxidative stress..."

Accordingly, we have added the appropriate references to substantiate this statement and carefully reviewed the manuscript to ensure that all claims are now supported by relevant citations or clearly qualified where evidence remains limited.

Referee #3

In this manuscript the authors extensively review of microplastics, persistent synthetic polymer particles, have emerged as an environmental hazard, notably affecting pulmonary health due to continuous inhalation exposure. The authors try to link the airborne microplastics and severe lung diseases, including cancer, asthma, COPD, and lung fibrosis, and provide the potential underlying mechanistic pathways. The review integrates emerging epidemiological evidence with recent mechanistic insights, highlighting how

inhaled microplastics disrupt lung cellular homeostasis by inducing inflammation, oxidative stress, ferroptosis, epithelial-mesenchymal transition, and epigenetic alterations.

This review is very timely and important. It is very comprehensive. My suggestions:

1. The manuscript is too long, more than 5,000 word counts and 132 references. It's may be more friendly to readers if the manuscript can be reduced to at least one-third or half.

We thank the reviewer for this helpful comment. We have made a concerted effort to streamline the manuscript by removing redundancies and improving readability, while also incorporating additional material requested by other reviewers to ensure completeness. Although the word count is still above 5000, we have focused on improving clarity and flow to make the text more accessible. We recognize that the reference list is extensive, but we chose to primarily cite original research articles rather than secondary reviews in order to appropriately credit the authors who generated the primary data.

2. A table to summarize the clinical relevance of each individual lung disease and microplastics exposure, the current epidemiologic evidence, mechanisms and related references. It will be even more readable.

In response, we have added a new table (see pages 33-36) summarizing the relevance of each major lung disease in relation to microplastic exposure, including the current epidemiological evidence, cellular response and key references.

3. In the introduction, "Recent studies demonstrate the systemic presence of microplastics, with particles detected in human blood, lung, heart, testes, and brain [3-7], raising significant concerns about bioaccumulation and disease risk." Presence of microplastics is not necessarily related to the diseases. Please extend.

We have revised the sentence to clarify that the detection of microplastics in human tissues does not imply causation. The revised text now emphasizes that while microplastics have been detected across multiple organ systems, the clinical relevance of this distribution remains uncertain.

4. In page 9, Are lung disorders altered by microplastics?

"Occupational studies have already documented adverse respiratory conditions among workers in plastic industries, such as those involved in PVC and nylon production. These studies link high concentrations of inhaled microplastics to serious lung diseases, including lung cancer." Please cite reference for this sentence.

The references have now been included.

30th Oct 2025

Dear Dr. Prekovic,

Thank you for submitting your revised manuscript to EMBO Molecular Medicine. We have now received feedback from referees 2 and 3 regarding your revisions. As you will see below, they recognise the effort made and consider the review to be improved. However, referee 2 still identifies areas that require further attention.

We would therefore like to invite you to make further revisions to address these remaining points. Please indicate in track changes mode any new modification to the text, and include the glossary in the manuscript text file.

Once you submit your revised manuscript, your figures will be sent to a graphic designer to be redrawn. Please define the different cell types in figure 2 or in its legend (for instance, for epithelial cells, there are different forms and colour cells) and ensure that all information in the figures is accurate.

Please also note:

- If the figure or single panels of the figure have been adapted from a published figure, please add this information to the figure legend.
- Please only re-use figures or parts of a figure if this is essential for understanding the concept communicated. If the figure contains re-used images or elements of images, please make sure that you have the permission/license to publish it. All re-used material must be explicitly cited.
- If you use an image data base for scientific iconography (e.g., BioRender), please let us know if you have a license that allows for publication in an academic journal.

Looking forward to receiving your revised manuscript,

With kind regards,

Lise Roth

Lise Roth, Ph.D.
Senior Editor
EMBO Molecular Medicine

***** Reviewer's comments *****

Referee #2 (Remarks for Author):

While some of the reviewer comments were addressed, not all issues were fixed. It is currently poorly written and detracts from the research topic. Overall, the manuscript needs to be thoroughly reviewed by the authors and written with more scientific rigor.

Manuscript still has extreme language (e.g. in intro - "vividly, dramatically, severe, demanding immediate scientific attention, Yet, identifying microplastics as active participants in disease progression presents an unprecedented opportunity - a decisive moment in public health where awareness and action can intersect. The emerging evidence demands more than recognition; it compels immediate scientific innovation and rigorous regulatory oversight etc..."). Manuscript needs to be revised to be written in scientific language format.

Manuscript still needs to be grammatically improved (e.g. in the intro, the following is not a sentence: Take the rapid industrialization of nineteenth-century London, for example.)

Please uniformly format references: (E.g., From properties to toxicity: Comparing microplastics to other airborne microparticles, 2022; Vasse & Melgert,

2024).

Some sentences are not clear and need to be revised: E.g., Their particulate nature combined with chemical complexity facilitates continuous interaction with and accumulation within biological systems, substantially enhancing their toxic potential.

Some sentences should be revised to be more scientific or eliminated: E.g., ...it is not inconceivable that their accumulation would be related to disease.

Please remove 90% of your "may" statements and focus on published research findings (e.g see page 12).

Please include a reference from this decade (...has shown notable shifts in clinical presentation and severity over recent decades (Kolb & Collard, 2014).

EMT not defined when using it for the first time.

Overstatement: What is notable is not merely the breadth of cellular disruption caused by microplastics, but their influence on disease trajectories themselves. Specifically, current literature suggests that microplastics may act as active modifiers that contribute to pathological processes underlying the aggressive phenotypes increasingly seen in lung cancer, COPD, asthma, and fibrosis

Referee #3 (Remarks for Author):

The authors have addressed the critiques and I have no further comment.

Referee #2 (Remarks for Author):

While some of the reviewer comments were addressed, not all issues were fixed. It is currently poorly written and detracts from the research topic. Overall, the manuscript needs to be thoroughly reviewed by the authors and written with more scientific rigor.

Manuscript still has extreme language (e.g. in intro - "vividly, dramatically, severe, demanding immediate scientific attention, Yet, identifying microplastics as active participants in disease progression presents an unprecedented opportunity - a decisive moment in public health where awareness and action can intersect. The emerging evidence demands more than recognition; it compels immediate scientific innovation and rigorous regulatory oversight etc..."). Manuscript needs to be revised to be written in scientific language format.

Manuscript still needs to be grammatically improved (e.g. in the intro, the following is not a sentence: Take the rapid industrialization of nineteenth-century London, for example.)

Please uniformly format references:

(E.g., From properties to toxicity: Comparing microplastics to other airborne microparticles, 2022; Vasse & Melgert, 2024).

Some sentences are not clear and need to be revised: E.g., Their particulate nature combined with chemical complexity facilitates continuous interaction with and accumulation within biological systems, substantially enhancing their toxic potential.

Some sentences should be revised to be more scientific or eliminated: E.g., ...it is not inconceivable that their accumulation would be related to disease.

Please remove 90% of your "may" statements and focus on published research findings (e.g see page 12).

Please include a reference from this decade (...has shown notable shifts in clinical presentation and severity over recent decades (Kolb & Collard, 2014).

EMT not defined when using it for the first time.

Overstatement: What is notable is not merely the breadth of cellular disruption caused by microplastics, but their influence on disease trajectories themselves. Specifically, current literature suggests that microplastics may act as active modifiers that contribute to pathological processes underlying the aggressive phenotypes increasingly seen in lung cancer, COPD, asthma, and fibrosis.

Author Response:

We thank the reviewer for their additional feedback and the opportunity to further refine our manuscript. We respectfully disagree with the statement that the manuscript is poorly written. The manuscript was originally prepared with careful attention to scientific accuracy, structure, and readability, and has been both written and critically reviewed by three native English speakers with experience in scientific publishing. The current version has been further refined to ensure full alignment with the stylistic and scientific standards of EMBO Molecular Medicine. All sections were reviewed line by line for clarity, coherence, and grammatical precision, ensuring that the manuscript communicates its scientific content with clarity and rigor.

To maintain a clear and consistent scientific tone, we carefully re-examined the manuscript for linguistic accuracy and precision. We removed or rephrased all remaining instances of language that could be interpreted as emphatic or editorial,

replacing them with neutral, evidence-based terminology. Words such as “vividly,” “dramatically,” “demanding immediate scientific attention,” and “decisive moment” have been replaced with phrasing consistent with standard scientific writing. The introductory paragraph, including the London industrialization example, has been rewritten into a complete and grammatically precise sentence that maintains historical context while adhering to formal academic style.

All references have been uniformly formatted according to *EMBO Molecular Medicine* guidelines and cross-checked for completeness and accuracy (for example, Vasse & Melgert, 2024; Wieland et al., 2022). We have also incorporated recent references from the last decade, such as Wolters et al., 2018, to support the statement regarding evolving clinical presentation and severity in pulmonary fibrosis. Sentences previously identified as ambiguous or speculative (for example, “it is not inconceivable that...”) have been rewritten to maintain precision and scientific objectivity. Similarly, any wording that could imply causality beyond the evidence available (for example, “active participants in disease progression” or “influence on disease trajectories”) has been revised to emphasize associations and mechanistic plausibility only, in accordance with current data.

We also addressed all technical and structural comments. The glossary has now been integrated into the main text file, and all abbreviations, including epithelial–mesenchymal transition (EMT), are defined upon first mention. Phrases and sentences highlighted as unclear have been rewritten to improve readability, including those describing particle behavior and cellular interaction mechanisms. Furthermore, the section on dose–response relationships has been revised for accuracy and precision, using language that reflects current limitations in the available data.

These collective changes strengthen the manuscript both scientifically and stylistically. The revised version conveys the complexity of the topic in a balanced and evidence-based manner, avoids overstated conclusions, and maintains clarity and flow throughout. We believe that the manuscript, already carefully prepared in its original form, now achieves an even higher standard of scientific rigor, readability, and stylistic consistency, and fully addresses the reviewer’s concerns.

20th Nov 2025

Dear Dr. Prekovic, Dear Stefan,

Thank you for submitting your revised manuscript. We have sent your figures to our graphic designer, who will redraw them and get in touch with you once a first draft is ready (usually within 10 days).

I am pleased to inform you that your manuscript is now accepted for publication and will be sent to our publisher once you approve the figures.

Your manuscript will be processed for publication by EMBO Press. It will be copy edited and you will receive page proofs prior to publication. Please note that you will be contacted by Springer Nature Author Services to complete licensing information.

This Review is free of charge. When you are contacted in a few weeks to sign your license agreement and review article proofs, please enter the following token into the relevant field in the Springer Nature author services system: [removed]

If you have any questions, please do not hesitate to contact the Editorial Office.
Thank you for your nice contribution to EMBO Molecular Medicine!

With kind regards,

Lise
